# Concurrent and Simultaneous Use of Cannabis and Tobacco and Its Relationship with Academic Achievement amongst University Students

**DOI:** 10.3390/bs8030031

**Published:** 2018-03-01

**Authors:** Olga Hernández-Serrano, Maria E. Gras, Sílvia Font-Mayolas

**Affiliations:** 1Departament of Physical Therapy, Campus of Salt, EUSES—University of Girona (UdG), Carrer Francesc Macià, 65, 17190 Salt, Girona, Spain; 2Department of Psychology, University of Girona (UdG), Plaça Sant Domènec, 9, 17004 Girona, Spain; eugenia.gras@udg.edu (M.E.G.); silvia.font@udg.edu (S.F.-M.)

**Keywords:** academic achievement, tobacco, cannabis, polydrug use, concurrent use, simultaneous use, university students

## Abstract

The combined use of cannabis and tobacco is frequent in Europe. Few studies have nonetheless explored this pattern of consumption and its relationship with academic achievement in Spanish population. The aim of the present study was to analyze (1) the frequency of four patterns of polydrug use the last year (non-dual users of cannabis and tobacco; concurrent users: cannabis and tobacco separately; simultaneous users: tobacco in cannabis “joints”; simultaneous users: tobacco in cannabis joints alongside alcohol) by gender and age; (2) grade point average (GPA) by gender and age; (3) the association between the frequency of the four patterns of use and the GPA amongst a sample of 477 Spanish university students. The use of cannabis and tobacco (concurrent and simultaneous) and GPA were assessed by means of self-reported questionnaires. Statistically significant differences were found for the GPA with respect to gender. The GPA by the non-dual users of cannabis and tobacco was significantly higher than the GPA corresponding to the concurrent and simultaneous users. The combined use of cannabis and tobacco, regardless of the type of use (concurrent or simultaneous), is moderately related to poor academic achievement amongst university students.

## 1. Introduction

Cannabis and tobacco are two common drugs often used in combination, especially “tobacco in cannabis joints”, in Europe [1,2]. Many of the studies performed in Spain do not distinguish between concurrent (i.e., separately smoking cannabis and tobacco within a given time period) and simultaneous (i.e., smoking cannabis in a “joint” or “blunt” with tobacco on a single occasion) polydrug use. Thus, researchers have generally ignored these patterns of use of cannabis and tobacco. This distinction is important since the use of cannabis mixed with tobacco, in comparison to smokeless forms, is more likely to develop cannabis dependence [3].

The co-use of cannabis and tobacco causes serious mental health and neurocognitive problems [4,5,6,7,8]. Academic performance is an important issue in our current society, since they prepare young people for their incorporation into working life. To date, most of the research conducted on university students has focused on analyzing the effects of cannabis or tobacco separately, with few studies focusing on their combined effects: the effects of the combined use of cannabis and tobacco on academic achievement may meaningfully influence the eventual success of future students.

The aim of the present study was to analyze (1) the frequency of four patterns of polydrug use in the last year (non-dual users of cannabis and tobacco; concurrent users: cannabis and tobacco separately; simultaneous users: tobacco in cannabis joints; simultaneous users: tobacco in cannabis joints alongside alcohol) by gender and age; (2) grade point average (GPA) by gender and age; (3) the association between the frequency of the four patterns of use and GPA amongst a sample of 477 Spanish university students.

## 2. Literature Review

### 2.1. Polydrug Use involving Cannabis and Tobacco

The World Health Organization currently defines the term “polydrug use” as the consumption of more than one kind of drug by an individual [9]. This consumption can be either concurrent (separately: the use of two or more substances within a given time period) or simultaneous (at the same time: the use of two or more substances on a single occasion) [10,11]. Cannabis and tobacco are commonly used drugs. In Europe, cannabis is more commonly consumed mixed with tobacco than isolated, in contrast to America [1,2]. According to the Spanish Observatory for Drugs and Toxicology [12], cannabis is consumed by approximately 60% of polydrug users. A vast majority (90.7%) of cannabis users in Spain aged 15–34 years old smoke cannabis mixed with tobacco. The Observatory also reports that the average consumption of joints is higher in boys than in girls [12]. Systematic reviews performed within this field highlight the need for further research exploring gender differences on the interaction between tobacco and marijuana, with the aim of elucidating the existing inconsistency across the results of different studies [13]. With regard to age, the average consumption in men scarcely varies with age, whereas in women, the number of joints consumed per day decreases with age [12]. Many studies have suggested that tobacco use increases the probability of developing dependency on cannabis and, at the same time, cannabis use increases the probability of large-scale tobacco use [14,15]. A recent study about cannabis and tobacco routes of administration by country found that consuming cannabis-mixed tobacco in joints (85.4%) is more frequent than consuming cannabis without tobacco in joints in Spain (7.9%) [2]. Moreover, the use of cannabis-mixed tobacco in joints (simultaneous use), in comparison to smokeless forms (i.e., concurrent use), is more likely to develop cannabis dependence [3].

Alcohol use is present across almost all patterns of polydrug use amongst young adults [16]. Alcohol and tobacco, followed by illicit drugs (especially marijuana), form a frequent pattern of drug combination amongst young adults [17,18,19] and college students [20,21,22] for both genders [22].

### 2.2. Cannabis and Tobacco Use and Its Relationship with Academic Achievement

Cannabis is the most commonly used illicit drug amongst university students [23]. Its potential detrimental effects on cognitive functions that affect academic performance (i.e., attention, concentration, memory, verbal fluency, processing speed, planning, and decision making) are of particular concern [24,25,26,27,28], since cannabis use is also directly and negatively associated with academic performance (i.e., dropping out of college, poorer performance on exams, less time studying, lower class attendance, and the likelihood of earning a college degree) [29,30,31,32,33]. Suerken et al. [34] found that all marijuana user groups (infrequent users, decreasing users, increasing users, and frequent users) reported lower average academic outcomes than non-users. In this sense, some studies have shown that smoking cigarettes is strongly associated with subsequent adverse educational outcomes [35,36]. Students with high academic performance tend to be nondaily tobacco users [37]. Similar results have been found when polydrug users of tobacco, cannabis and alcohol were studied. For example, Mazur et al. [38] found that girls with a poor perception of school performance, compared to girls with better achievements, were at higher risk of being polydrug users of tobacco, cannabis, and alcohol (considering drug use across the last 30 days). In contrast to previous literature, a recent placebo-controlled trial has found that tobacco improves working memory across all load levels including maintenance, manipulation, and attention, independently of cannabis. This study concludes that tobacco may offset the effects of cannabis on delayed verbal recall [39].

Most college students successfully cope with a complex new life role and subsequently achieve academic success, but other students are less able to manage their studies successfully. In Spain, the number of students enrolled in university degrees decreased in 127,674 individuals between the academic years 2011–2012 and 2015–2016. At the same time, the rate of dropouts during the first year of university studies has increased from a percentage of 19% during the academic year 2011–2012 to 22.5% in 2015–2016 [40]. In this sense, prior studies have established an association between college academic achievement and retention, with higher-performing students persisting in their studies to a greater extent than lower-achieving students [41,42]. Furthermore, academic performance in Spanish female students is higher than in male students across the continuum of learning, which entails a higher female representation in the university system in Spain as of today [40]. However, studies have shown that the relationship between gender and academic achievement could be linked to the branch of knowledge assessed [43]: male students tend to show better academic results in scientific and technological studies, whereas female students usually perform better in health and well-fare university studies [44].

## 3. Materials and Methods

### 3.1. Participants

Participants were students in physical therapy (66.2%, n = 316) and sports science (33.8%, n = 161) from a university in Girona, Catalonia, Spain. The center was selected by means of a nonprobability purposive sampling method. The sample was composed entirely of students (100%) from compulsory-attendance classes, from first, second, and third year studies, taught by lecturers collaborating in the research. A total of 477 university students (56.2% female, 43.6% male, 0.2% non-reported) 17–40 years of age (21.33 ± 2.81) were enrolled in the study.

### 3.2. Instruments

Sociodemographic variables (degree, year of studies, age, and gender) were assessed by means of an ad hoc questionnaire.

Data on drug use were gathered through a self-report assessing the frequency of consumption during the last year. Individuals were asked to respond to three questions on a 5-point Likert scale (“never”, “occasionally”, “once a week”, “more than once a week”, and “daily”). The questions focused on when, during the last year, they have smoked (1) cannabis and tobacco separately, (2) cannabis mixed with tobacco in joints, and (3) cannabis mixed with tobacco in joints alongside alcohol. Participants reporting occasional or more frequent consumption were considered as “consumers”. Four patterns were therefore defined: students avoiding mainly cannabis and tobacco (non-dual users: CAN-TOB); students smoking cannabis and tobacco separately in the last year (concurrent users: CAN-TOB); students smoking cannabis with tobacco at the same time on a single occasion, “tobacco in cannabis joints” (simultaneous users: CAN-TOB); students smoking cannabis with tobacco at the same time on a single occasion, “tobacco in cannabis joints”, alongside alcohol (simultaneous users: CAN-TOB-ALC).

The variable “academic achievement” (GPA) was assessed by means of a self-report composed of an item focusing on the average mark obtained during the previous semester, with quantitative response values from 0 to 10 points (with the possibility of introducing intermediate values). The specific question assessed was, “based on the marks obtained in each subject during the past semester (academic period encompassing from 1 September 2014 to 1 February 2015, and taking into account both the pass marks (from 5 to 10 points) and the non-pass marks (from 0 to 4.9 points) obtained, what would be your average mark for the past semester?”.

### 3.3. Procedure

The present study was approved by the Research Committee of the University of Girona (UdG). Prior to data collection, an interview with the managing staff of the university school was conducted, with the aim of expounding on the main characteristics of the research and to request participation. The request for participation from students was carried out by specific written consent. All students accepted participation voluntarily after being informed on the purpose of the study and the respect to the ethical principles in research. Participants provided answers to the questionnaires within lecture-rooms of the university during the second semester, in February 2015 (see Figure 1).

### 3.4. Statistical Analysis

A chi-square test was performed to compare the amount of consumers by gender, whereas an ANOVA test was used to analyze consumers by age. The relationship between academic achievement and gender was tested by means of a T-test, whilst Pearson correlation analysis was used to explore the relationship between academic achievement and age. Finally, two-factor (gender/consumption) ANOVA was used to explore the average mark in academic achievement. The effect of the pattern of over GPA was analyzed with the size of the effect (η^2^). SPSS version 19.0 (IBM, Armonk, NY, USA) was used throughout the analyses performed.

## 4. Results

### 4.1. Patterns of Polydrug Use (Non-Dual Use: CAN-TOB, Concurrent Use: CAN-TOB, Simultaneous Use: CAN-TOB and CAN-TOB-ALC) by Gender and Age

Table 1 shows higher percentages of concurrent users (CAN-TOB) amongst female students with respect to the male group. Conversely, male students showed higher rates of simultaneous use (CAN-TOB and CAN-TOB-ALC) than female students.

Table 2 shows higher mean values in age in simultaneous users (CAN-TOB and CAN-TOB-ALC) than in concurrent users (CAN-TOB). However, no statistically significant differences in the mean age of the four different groups were found.

### 4.2. Academic Achievement (GPA) by Gender and Age

The average mark obtained during the last semester showed statistically significant differences between male and female students (*t* = 2.68; *p* = 0.008). The average mark obtained was higher in women (6.75 ± 0.83) than in men (6.51 ± 0.99).

According to the results of Pearson correlation, there was no relationship between academic achievement and age (*r* = 0.022; *p* = 0.64).

### 4.3. Academic Achievement (GPA) by Cannabis and Tobacco Use (Non-Dual Use: CAN-TOB; Concurrent Use: CAN-TOB; Simultaneous Use: CAN-TOB and CAN-TOB-ALC) and Gender

Table 3 shows the GPA by patterns of polydrug use (non-dual use: CAN-TOB; concurrent use: CAN-TOB; simultaneous use: CAN-TOB and CAN-TOB-ALC) and gender. The interaction effect between gender and pattern of consumption showed no significant differences (*F* = 0.737; *p* = 0.53).

The effect of the pattern of consumption over GPA was statistically significant (*F* = 6.26; *p* < 0.001) with a medium effect size (η^2^ = 0.041). Thus, the average mark of the students with a non-dual use pattern was higher than the GPA from polydrug users: concurrent users: CAN-TOB (*p* = 0.005); simultaneous users: CAN-TOB (*p* = 0.023); simultaneous users: CAN-TOB-ALC (*p* = 0.001).

Finally, no significant differences were found amongst the three patterns of polydrug users (concurrent users: CAN-TOB; simultaneous users: CAN-TOB and CAN-TOB-ALC) in relation to academic achievement.

## 5. Discussion

The first purpose of this study was to analyze the frequency of four different patterns of polydrug use (non-dual use: CAN-TOB; concurrent use: CAN-TOB; simultaneous use: CAN-TOB and CAN-TOB-ALC), with respect to gender and age. The percentages of non-dual consumer (CAN-TOB) and polydrug users (concurrent users: CAN-TOB; simultaneous users: CAN-TOB and CAN-TOB-ALC) did not significantly differ amongst male and female students. This result is consistent with previous research assessing polydrug use (consumers of cannabis plus alcohol and/or tobacco consumers) in university students, since no differences by gender were found [22]. However, results stemming from other studies confirm that the co-use of cannabis and tobacco is more frequent in men with respect to women [12,45,46]. Several explanations on the lack of differences between the performances of men and women arise: firstly, the results of studies on cigarette smoking among European students have shown that the average lifetime prevalence of cigarette smoking is similar in boys (47%) and girls (44%) [1]. Parallel results are found in Spanish national surveys, in which the prevalence of tobacco use remained steady from 2011 until the last survey conducted in 2015 [12]. Moreover, the perception of risk associated with the use of cannabis, either sporadically or ordinarily, has decreased in the population between 15 and 64 year of age [12]. Despite the fact that cannabis is more frequently consumed amongst men than by women [1,12], the steady tendency on tobacco in the last year, along with the decrease in the perception of risk, could be influencing the tendency to balance the consumption of cannabis and tobacco, with or without alcohol, in both genders. Other alternative explanations could stem from the fact that the present study did not differentiate the use of marijuana + tobacco from the duality hashish + tobacco. According to the Spanish Observatory of Drugs and Drug Addiction, marijuana is more widespread than hashish among cannabis users (men and women) in Spain [12]. Thus, the lack of differences between men and women in the form of consumption (concurrent use or simultaneous use) might be explained by a higher consumption of marijuana than hashish in both sexes. Further studies are nonetheless needed in order to clarify the relationship between polydrug use and gender.

Although no significant differences were found concerning the mean age of participants across the three groups of consumption (concurrent users: CAN-TOB; simultaneous users: CAN-TOB; simultaneous users: CAN-TOB-ALC), the mean age of the simultaneous user group (CAN-TOB-ALC) was higher with respect to the two extant cannabis and tobacco user groups (concurrent users: CAN-TOB; simultaneous users: CAN-TOB). Previous studies highlight an increase in the simultaneous polydrug use of various substances correlating with the progression in the year of studies [47]. In Europe, as in Spain, cannabis is an illicit drug with higher rates of consumption across all different age groups, even though the higher rate of users corresponds to the 15-to-24-year-old group [1,12]. The mean age of the first consumption of cannabis is 16 years of age, whereas 25 years is the mean age at which treatment for troubles derived from the use of cannabis is initiated [1]. Considering the fact that, in the present study, the mean age for initiating the combined use of cannabis and tobacco is 21 years, the development of further longitudinal studies could provide information on the change from cannabis use to polydrug use (concurrent and/or simultaneous use). This fact could lead to the development of early tailored interventions focusing on the prevention of the combined use of both substances (concurrent and simultaneous use of cannabis and tobacco).

Results related to the second purpose of the present study reveal that the GPA among male students is lower than in females. This result is consistent and endorsed by the fact that more women (53.1%) than men (46.9%) have completed university education studies in Spain [40]. Our data are also consistent with previous research conducted among the adolescent population [37,48]. Notwithstanding, as stated above, these differences have been linked to the different characteristics of the university degrees (scientific-technological vs health-related). Female students usually show better academic results than male individuals within the field of health-related studies [43,44], as in the case at hand. On another note, GPA did not significantly differ by age. Based on the most recent data published by the Ministry of Education, Culture and Sports in Spain [40], the drop-out rates in Spanish universities regarding the first and second year of studies have increased, but, despite this fact, the students that remain and pursue their university studies maintain a more stable academic degree of performance throughout the academic years with respect to previous years (2011/2012). Since the present study has not assessed the percentage of drop-outs, no relationship is susceptible to be explored in this sense: further research should take this aspect into account.

The third purpose of this study was to analyze the association between the frequency of four patterns of polydrug use (non-dual use: CAN-TOB; concurrent use: CAN-TOB; simultaneous use: CAN-TOB and CAN-TOB-ALC) and academic achievement (GPA). The three cannabis and tobacco user groups (concurrent use: CAN-TOB; simultaneous use: CAN-TOB and CAN-TOB-ALC) reported lower GPAs, on average, than the non-dual users. There are at least three potential interpretations for this result: (1) The concurrent and simultaneous use of cannabis with tobacco could cause poor GPAs. In this sense, Hindocha et al. in 2017 [39] examined the individual and interactive effects of cannabis and tobacco. The results showed that cannabis alone impaired verbal and working memory, whilst tobacco alone improved the working memory. When tobacco was combined with cannabis, the impairment in delayed recall was attenuated with respect to the use of cannabis alone. This finding is interpreted as evidence that co-administration interferes with learning via cognitive functions. Other recent research focusing on the study of working memory under conditions of the simultaneous use of marijuana and tobacco showed that the putative effect of marijuana on working memory and the facilitative effect of tobacco on working memory were no longer present when used simultaneously with tobacco and alcohol, respectively [49]. Acute memory is needed to function and succeed in a university environment. However, a recent investigation concluded that the complex actions of polydrug use (concurrent and simultaneous use) on the brain structure and function need greater examination [50]. (2) Concurrent and simultaneous use of cannabis and tobacco could be a consequence of poor GPA. For example, in relation to concurrent use, cannabis users who also smoke tobacco are more dependent on cannabis, have more psychosocial problems, and have poorer cessation outcomes than those who use cannabis but not tobacco [51]. On the other hand, focusing on simultaneous use, studies have found an association between the number of drugs simultaneously used and social issues (i.e., social consequences and aggressiveness) [52]. (3) The relationship between the use of cannabis plus tobacco (either in a concurrent or simultaneous way) and poor levels of GPA may be explained by the presence of a third variable, such as affluent families [53], family dysfunctions [54], or parental education [55]. Further and more specific research on these eventual third variables is nonetheless needed, since the studies conducted to date amongst the Spanish population on consumption and GPA did not differentiate between cannabis use versus cannabis and tobacco and, moreover, no distinction was made between concurrent use of cannabis and tobacco versus simultaneous use of cannabis and tobacco.

Our findings must be interpreted in light of study limitations, which need to be solved in future research: (1) The sample corresponds to a specific geographical region and area of knowledge or study (university students in health and sport), a fact that hinders an eventual generalization to other contexts. Further studies with larger sample sizes and from different geographical areas shall be conducted. (2) No temporal relationships have been explored, since the present study is cross-sectional. A longitudinal study observing changes over time would be required to complete this research. (3) Given that the consumption of drugs may or may not be generally accepted by society, the answers given by the participants in the self-report could be inflated or deflated by a social desirability bias. The use of biological tests for assessing polydrug use and the inclusion of different informants (peers and/or parents) would provide more reliable measures of those variables. Nevertheless, confidentiality of responses was guaranteed in an effort to minimize this bias. (4) Our instruments to quantify concurrent and simultaneous use (CAN-TOB and CAN-TOB-ALC) did not measure grams of the psychoactive compounds. Systematic reviews and other studies concluded that a standard cannabis unit should be used in order to improve the assessment of cannabis use [56,57]. (5) Retrospective and self-report questionnaires are subject to recall bias and underreporting. Objective measures are required to evaluate the GPA in future studies. (6) The present study did not consider whether the non-dual user group could have consumed alcohol, other forms of nicotine or THC, and other drugs in the last year. Future research on the relationship between the use of cannabis and tobacco and academic achievement should consider the role of other drugs. (7) No distinction on the concurrent and simultaneous use of marijuana and tobacco versus hashish and tobacco was considered. Taking the findings of Spanish reports on the gender differences concerning the use of both substances into account [12], future research shall distinguish and specifically differentiate both substances.

## 6. Conclusions

This study examined the relationship between the polydrug use of cannabis with tobacco—concurrently (CAN-TOB) and simultaneously (CAN-TOB and CAN-TOB-ALC) and academic achievement. The combined use of cannabis and tobacco, regardless of the type of use (concurrent or simultaneous), is moderately related to poor academic achievement amongst university students. The design of future programs on cannabis prevention focusing on university students should include sessions centered in the combined use of cannabis and tobacco, with the aim of assessing their impact on academic achievement.

## Figures and Tables

**Figure 1 behavsci-08-00031-f001:**
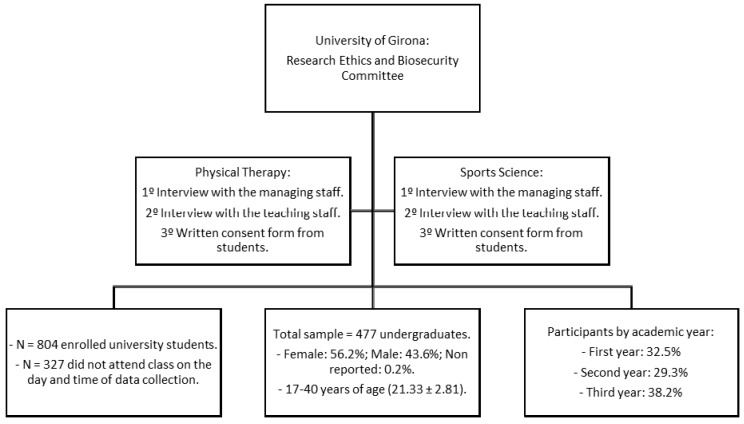
Procedure.

**Table 1 behavsci-08-00031-t001:** Patterns of polydrug use by gender.

Polydrug Use	Male (%)	Female (%)	χ^2^ (*p*)
Non-dual use: CAN-TOB	55.6	54.8	0.85 (0.84)
Concurrent use: CAN-TOB	32.8	35.1	
Simultaneous use: CAN-TOB	3.7	2.4	
Simultaneous use: CAN-TOB-ALC	7.8	7.7	

CAN: cannabis; TOB: tobacco; ALC: alcohol; *p* < 0.05.

**Table 2 behavsci-08-00031-t002:** Patterns of polydrug use by age.

Polydrug Use	Average Age	SD	F-Test (*p*)
Non dual use: CAN-TOB	21.3	2.89	1.12 (0.34)
Concurrent use: CAN-TOB	21.5	2.74	
Simultaneous use: CAN-TOB	21.6	2.35	
Simultaneous use: CAN-TOB-ALC	21.8	2.65	

CAN: cannabis; TOB: tobacco; ALC: alcohol; SD: standard deviation; *p* < 0.05.

**Table 3 behavsci-08-00031-t003:** GPA by gender and patterns of polydrug use.

	Male (n = 259 ^1^)			Female (n = 186 ^1^)			Total (n = 445 ^1^)		
Polydrug use	GPA	SD	N	GPA	SD	N	GPA	SD	N
Non-dual use: CAN-TOB	6.69	0.78	143	6.87	0.84	102	6.77	0.81	245
Concurrent use: CAN-TOB	6.38	1.16	85	6.64	0.80	66	6.50	1.02	151
Simultaneous use: CAN-TOB	6.19	1.46	10	6.12	0.25	4	6.17	1.22	14
Simultaneous use: CAN-TOB-ALC	5.91	0.94	21	6.54	0.82	14	6.16	0.93	35

CAN: cannabis; TOB: tobacco; ALC: alcohol; GPA: grade point average; SD: standard deviation; ^1^ 32 participants did not report their GPA.

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
