# Peer review of "Concurrent and Simultaneous Use of Cannabis and Tobacco and Its Relationship with Academic Achievement amongst University Students"

_behavsci, 2018, doi:10.3390/bs8030031_

Round 1

Reviewer 1 Report

The paper is relevant and includes information that has not been analyzed previously in Spanish university students. However, some considerations could improve paper readability:

- Author/s name as simultaneous consumers those who consume tobacco and cannabis with and without alcohol. A specific nomenclature for the alcohol consumption group would make easier to understand differences between groups. 

- Academic performance is commonly used in singular (i.e., academic performance is) rather than in plural, as appears in the paper (line 41). 

-In line 49 in my opinion an “and” is missed before “(3)”.

- Author/s state "explanations for the negative association challenge this interpretation". However, I do not consider it is a challenge but a different explanation of the same result. 

In "results" section, information regarding the relationship between GPA and age should be rewritten as it is referred to Pearson correlation and not to mean differences. 

In lines 184-185, author/s state "The effect of the pattern of consumption was statistically significant (F = 6.26; p < .001) with a medium effect size (eta-squared = 0.041)". It is necessary to include that the significance is referred to the relation between pattern of consumption and GPA, otherwise it could be more difficult to be understood. Furthermore, as the information regarding GPA and gender has been informed in the previous section (line 173-175), although congruent it is not necessary to be included. 

- In line 308 authors state that  “type of use (concurrent or simultaneous) is strongly related (...)". It should be changed by “moderately” according to the data. 

Author Response

Thank you for the invitation to revise and resubmit our manuscript entitled “Concurrent and Simultaneous Polydrug Use involving Cannabis and Tobacco and its relationship with the Academic Achievement amongst University Students” (behavsci-262351). As authors of this paper, we specially appreciate having received suggestions that will contribute to substantially improve the quality of our manuscript. We believe that the latest version of this paper is much improved by addressing the concerns raised by the reviewers.

Please find a detailed description on how your concerns have been addressed in the manuscript. We hope that you will continue to consider it to be of interest to the readers of Behavioral Sciences.

1. Author/s name as simultaneous consumers those who consume tobacco and cannabis with and without alcohol. A specific nomenclature for the alcohol consumption group would make easier to understand differences between groups.

Thank you for the indication provided: following your advice, we have added a specific nomenclature for the alcohol consumption group and the other consumption groups (please see pages 3-9, lines 140-345).  

2. Academic performance is commonly used in singular (i.e., academic performance is) rather than in plural, as appears in the paper (line 41).

Thank you very much for your accurate indication: we have included the singular term in all the sections of the article (lines 41 and 95).

3. In line 49 in my opinion an “and” is missed before “(3)”.

Thank you very much for your opinion. We have added "and" in the sections ‘abstract’ and ‘introduction’ (lines 22 and 51, respectively).

4. Author/s state "explanations for the negative association challenge this interpretation". However, I do not consider it is a challenge but a different explanation of the same result.

Thank you very much for your comment. The incorrect expression has been removed from the text.

5. In "results" section, information regarding the relationship between GPA and age should be rewritten as it is referred to Pearson correlation and not to mean differences.

We have replaced the sentence "No significant differences were found between the average mark and the mean age of students (r = 0.022; p = .64)" for the sentence "According to the results of Pearson correlation, there was no relationship between the academic achievement and age (r = 0.022; p = .64)".

6. In lines 184-185, author/s state "The effect of the pattern of consumption was statistically significant (F = 6.26; p < .001) with a medium effect size (eta-squared = 0.041)". It is necessary to include that the significance is referred to the relation between pattern of consumption and GPA, otherwise it could be more difficult to be understood. Furthermore, as the information regarding GPA and gender has been informed in the previous section (line 173-175), although congruent it is not necessary to be included.

We have clarified the sentence pointed-out (please see line 204). According to the indications of the reviewer, we have removed the sentence "the effect of gender to a similar performance, showing no differences either (F = 2.35; p <.13)".

7. In line 308 authors state that “type of use (concurrent or simultaneous) is strongly related (...)". It should be changed by “moderately” according to the data.

Thank you very much for your accurate indication: we have replaced “strongly” by “moderately”.

Thank you very much for your feedback on our article: we really hope that you’ll find our manuscript interesting and improved after the inclusion of the aspects stated and pointed out by the reviewers.

Reviewer 2 Report

    The paper presents a study on social factors for students’ assessment, which different from the previous single influence of cannabis or tobacco, attempts to explore their combined effects. The paper is in general well written.

    In order to correct the typos and make sure the expression is smooth; I suggest the authors asking the native English editors to revise this manuscript before publication.

    For facilitating readers easily read and quickly understand the related context of this study, it is recommended to do suitable segmentations of the section 2 Literature Review, and give subtitle.

    Currently, description of 3.3 Procedure is too rough, I suggest the authors to draw a flow chart with the entire experimental process in a sequential manner introduced, and more clearly present the experiment steps.

    It is recommended to keep the key words consistent such as the order of "Cannabis and Tobacco". Line 40, line 44, line 93, line 124, line 178, line 194,line 197, line 222, line 252, line 254, and the table 3, show the order "Tobacco and cannabis". In addition, mentioned many times in this study the different kinds of cannabis products such as: cannabis, marijuana, hashish, if having differences then should be defined.

    Most previous studies have made the general public aware that the harmful effects of cannabis and tobacco more than its benefits. As noted in line 40-41 and line 73-86 of this article, the use of cannabis and tobacco to users health and cognition have negative impacts, and the literature presented in this article the negative more than positive, readers are expected to form: (1) non-users are better at academic achievement than those uses of cannabis and tobacco, and (2) students who are in more complex use situation, the high probability of worse academic performance they will have. I suggest the authors add references to balance too much negative opinions and clarify the facts from contradictions through experiments to highlight the contribution of this research.

Author Response

Thank you for the invitation to revise and resubmit our manuscript entitled “Concurrent and Simultaneous Polydrug Use involving Cannabis and Tobacco and its relationship with the Academic Achievement amongst University Students” (behavsci-262351). As authors of this paper, we specially appreciate having received suggestions that will contribute to substantially improve the quality of our manuscript. We believe that the latest version of this paper is much improved by addressing the concerns raised by the reviewers.

Please find a detailed description on how your concerns have been addressed in the manuscript. We hope that you will continue to consider it to be of interest to the readers of Behavioral Sciences.

1. In order to correct the typos and make sure the expression is smooth; I suggest the authors asking the native English editors to revise this manuscript before publication.

According to the information included in the acknowledgments section, the document was previously reviewed by a native English editor (Dr. M. G.). The manuscript has nonetheless been reviewed again by the same native English editor (Dr. M. G.) and some minor formal changes have been subsequently implemented.

2. For facilitating readers easily read and quickly understand the related context of this study, it is recommended to do suitable segmentations of the section 2 Literature Review, and give subtitle.

Thank you for your recommendation. We have done segmentations of the section 2 Literature Review.

3. Currently, description of 3.3 Procedure is too rough, I suggest the authors to draw a flow chart with the entire experimental process in a sequential manner introduced, and more clearly present the experiment steps.

We have clarified the experiment steps in the Procedure section by means of a flow chart (figure 1): thank you for your recommendation.

4. It is recommended to keep the key words consistent such as the order of "Cannabis and Tobacco". Line 40, line 44, line 93, line 124, line 178, line 194,line 197, line 222, line 252, line 254, and the table 3, show the order "Tobacco and cannabis". In addition, mentioned many times in this study the different kinds of cannabis products such as: cannabis, marijuana, hashish, if having differences then should be defined.

We have changed the order of "Cannabis and Tobacco" in all sections of the manuscript. In addition, we have defined the differences between the types of cannabis products (page 7, lines 241-249).

5. Most previous studies have made the general public aware that the harmful effects of cannabis and tobacco more than its benefits. As noted in line 40-41 and line 73-86 of this article, the use of cannabis and tobacco to users health and cognition have negative impacts, and the literature presented in this article the negative more than positive, readers are expected to form: (1) non-users are better at academic achievement than those uses of cannabis and tobacco, and (2) students who are in more complex use situation, the high probability of worse academic performance they will have. I suggest the authors add references to balance too much negative opinions and clarify the facts from contradictions through experiments to highlight the contribution of this research.

We have added references to balance the negative opinions and to clarify the facts from contradictions. Thank you for your recommendation.

Thank you very much for your feedback on our article: we really hope that you’ll find our manuscript interesting and improved after the inclusion of the aspects stated and pointed out by the reviewers.

Reviewer 3 Report

Comments to the Author

Review of Concurrent and Simultaneous Polydrug Use Involving Cannabis and Tobacco and Its Relationships with Academic Achievement Amongst University Students

This is an interesting report investigating the impact of polydrug use on GPA among university students in Spain. The overall conclusions of this report are that polydrug use, regardless of simultaneous or concurrent use, negatively impacts GPA. While this topic is important for considerations, the information could be delivered more succinctly and the arguments made more strongly after altering some methodological approaches and making this manuscript into a brief report. Below are listed some general comments and recommendations that have led to the recommendation of a brief report with major revisions.

General Comments and Recommendations:

Beyond shortening the overall manuscript and correcting for typos and complicated wording, here are several methodological suggestions for improvement:

A major methodological issue that is of concern is the definition of “non-users” in this study. According to the Instruments section, line 127-128, a non-user is defined as someone who checks never to smoking tobacco AND cannabis. The way the question is phrased, or perhaps only the way it is reported in the manuscript, seems to indicate that if someone only smoked tobacco or only smoked cannabis, they would be considered a non-user. A more accurate term, which would also change the conclusions of this paper, would be non-dual users. If non-user is defined differently, that needs to be clarified in this section.

Another area of improvement that would increase the quality of analyses is that single use was not included in these analyses. A more useful comparison may be between those who only use cannabis compared to combined users. If this information is available in your dataset, I would recommend re-doing analyses to include the comparison to single drug users.

Finally, frequency of use may also be an important factor to consider, such as comparing less frequent users to more frequent/ daily users. If this factor was examined and no significant differences were found, or frequency did not impact that outcomes of the analyses, it should be addressed in the paper.

Literature Review Section Recommendations:

1.     Remove longitudinal background information in your literature review section if you are not going to address temporal effects in your paper (approximately lines 82-89).

2.     Similarly, remove information about factors that impact use and GPA if you are not going to address them in this study, (e.g. impact of family functioning and socioeconomic status, lines 90-94). I’d also recommend excluding information about dropout rates if you don’t examine this in the study.

3.     The final paragraph in the literature review section needs to be tied back to the current study. The discussion of academics is relevant but the connection between this section and the exploration of poly-drug use by gender could be made more explicit.

4.     Finally, the rationale for the inclusion of alcohol use should be included in the introduction and literature review or the simultaneous category including alcohol used needs to be removed from analyses. It is unclear why this is an important distinction and what part of your hypotheses is answered by adding alcohol to these analyses.

Results Section Recommendations:

5.     Clarify sentence on line 163 (perhaps by removing nonetheless) so it is clear that the comparisons were not significant.

Discussion Section Recommendations:

6.     Speak to why gender did not remain significant in the analyses examining gender and poly-drug use (Section 4.3).

7.     It is unclear what important conclusions are to be drawn from the paragraph including lines 204-217. Please clarify wording and tie it back to the results found specifically in this study.

8.     Lines 267-268 have “new” information that wasn’t clearly laid out in the results. Include more about this finding in the results section.

9.     On line 299, when discussing that there is no data on whether non-users used alcohol, there should also be statements about the potential use of other drugs or other forms of nicotine or THC.

10.  Methodological limitations that need to be addressed in discussion:

-       A convenience sample from two very specific sections of a University. Since these programs are health-focused, it may not be a representative sample. Further, based on how it is written, it seems that only first and third year students were included in the survey, missing out on a wider range of students.

-       The potential validity issues from self-reporting substance use without biochemical verification and self-reporting GPA.

Author Response

Thank you for the invitation to revise and resubmit our manuscript entitled “Concurrent and Simultaneous Polydrug Use involving Cannabis and Tobacco and its relationship with the Academic Achievement amongst University Students” (behavsci-262351). As authors of this paper, we specially appreciate having received suggestions that will contribute to substantially improve the quality of our manuscript. We believe that the latest version of this paper is much improved by addressing the concerns raised by the reviewers.

Please find a detailed description on how your concerns have been addressed in the manuscript. We hope that you will continue to consider it to be of interest to the readers of Behavioral Sciences.

General Comments and Recommendations:

1. A major methodological issue that is of concern is the definition of “non-users” in this study. According to the Instruments section, line 127-128, a non-user is defined as someone who checks never to smoking tobacco AND cannabis. The way the question is phrased, or perhaps only the way it is reported in the manuscript, seems to indicate that if someone only smoked tobacco or only smoked cannabis, they would be considered a non-user. A more accurate term, which would also change the conclusions of this paper, would be non-dual users. If non-user is defined differently, that needs to be clarified in this section.

The authors appreciate your suggestions for improvement. We have changed "non-user" by "non-dual users" across all the sections of the article.

2. Another area of improvement that would increase the quality of analyses is that single use was not included in these analyses. A more useful comparison may be between those who only use cannabis compared to combined users. If this information is available in your dataset, I would recommend re-doing analyses to include the comparison to single drug users.

3. Finally, frequency of use may also be an important factor to consider, such as comparing less frequent users to more frequent/ daily users. If this factor was examined and no significant differences were found, or frequency did not impact that outcomes of the analyses, it should be addressed in the paper.

The authors understand that the comparison between cannabis users and polydrug users, as well as comparing less frequent users versus more frequent / daily users, could be useful. However, the information about the comparison between cannabis users and polydrug users was previously used in another research article. On the other hand, the information about the comparison between "less frequent consumers" and "more frequent consumers/ daily users" was not included in this article because "more frequent consumers" corresponded to a small percentage. This small percentage did not allow for powerful statistical tests. Thank you for your recommendations: we will consider your comments for future studies.

Literature Review Section Recommendations:

4. Remove longitudinal background information in your literature review section if you are not going to address temporal effects in your paper (approximately lines 82-89).

Thank you indeed for your recommendation. We have removed the information about the longitudinal background (article: 31, 32 and 33).

5. Similarly, remove information about factors that impact use and GPA if you are not going to address them in this study, (e.g. impact of family functioning and socioeconomic status, lines 90-94). I’d also recommend excluding information about dropout rates if you don’t examine this in the study.

We have removed the information about the impact of family functioning and dropout rates (article 37, 39 and 40).

6. The final paragraph in the literature review section needs to be tied back to the current study. The discussion of academics is relevant but the connection between this section and the exploration of poly-drug use by gender could be made more explicit.

We appreciate your suggestion: we have modified this paragraph (please see the final paragraph in the literature review).

7. Finally, the rationale for the inclusion of alcohol use should be included in the introduction and literature review or the simultaneous category including alcohol used needs to be removed from analyses. It is unclear why this is an important distinction and what part of your hypotheses is answered by adding alcohol to these analyses.

We have included information about the rationale for the inclusion of alcohol use (see page 2, line 75-78).

Results Section Recommendations:

8. Clarify sentence on line 163 (perhaps by removing nonetheless) so it is clear that the comparisons were not significant.

We have removed this sentence. Thank you for the recommendation.

Discussion Section Recommendations:

9. Speak to why gender did not remain significant in the analyses examining gender and poly-drug use (Section 4.3).

The first paragraph of the discussion section explains and develops why gender and polydrug use did not significantly differ between male and female students. In this sense, we have included changes to improve our explanations (see page 7, line 234-249)

10. It is unclear what important conclusions are to be drawn from the paragraph including lines 204-217. Please clarify wording and tie it back to the results found specifically in this study.

We have clarified the conclusions in this paragraph. Thank you.

11. Lines 267-268 have “new” information that wasn’t clearly laid out in the results. Include more about this finding in the results section.

Thank you very much for your comment. This sentence presented an incorrect expression that we have removed.

12. On line 299, when discussing that there is no data on whether non-users used alcohol, there should also be statements about the potential use of other drugs or other forms of nicotine or THC.

Thank you very much for your recommendation. We have included information about the use of other drugs and other forms of nicotine and THC (please see page 9, line 339).

13. Methodological limitations that need to be addressed in discussion:

- A convenience sample from two very specific sections of a University. Since these programs are health-focused, it may not be a representative sample. Further, based on how it is written, it seems that only first and third year students were included in the survey, missing out on a wider range of students.

- The potential validity issues from self-reporting substance use without biochemical verification and self-reporting GPA.

This is correct: the sample was about a specific area of knowledge or study (health and sport area) and it could be a limitation. We have included this information in the discussion section (limitations, lines 323-324) besides the information about the potential validity issues from self-reporting substance use without biochemical verification and self-reporting GPA (lines 336-337).

In relation to the year of study, students from first, second and third years of study were included. We have improved the explanation concerning this information (line 127 and figure 1). Thank you very much for the suggestions provided.

Thank you very much for your feedback on our article: we really hope that you’ll find our manuscript interesting and improved after the inclusion of the aspects stated and pointed out by the reviewers.

Round 2

Reviewer 2 Report

Generally, the authors have made improvements to my recommendations and improved the quality of the articles.

Author Response

Thank you very much for your feedback on our article: we really hope that you’ll find our manuscript improved after the inclusion of the minor revisions and pointed out by the academic editor.

Reviewer 3 Report

Authors have addressed my concerns and I believe this revision is much improved.

Author Response

(The authors gave the same response as above.)
